# MoEMeta: Mixture-of-Experts Meta Learning for Few-Shot Relational Learning

**Han Wu[1,2], Jie Yin[1]**[*]
[1]The University of Sydney, Australia
[2]Peking University, China
{han.wu, jie.yin@sydney.edu.au}

## Abstract

Few-shot knowledge graph relational learning seeks to perform reasoning over relations given only a limited number of training examples. While existing approaches largely adopt a meta-learning framework for enabling fast adaptation to new relations, they suffer from two key pitfalls. First, they learn relation meta-knowledge in isolation, failing to capture common relational patterns shared across tasks. Second, they struggle to effectively incorporate local, task-specific contexts crucial for rapid adaptation. To address these limitations, we propose MoEMeta, a novel meta-learning framework that disentangles globally shared knowledge from task-specific contexts to enable both effective model generalization and rapid adaptation. MoEMeta introduces two key innovations: (i) a mixture-of-experts (MoE) model that learns globally shared relational prototypes to enhance generalization, and (ii) a task-tailored adaptation mechanism that captures local contexts for fast task-specific adaptation. By balancing global generalization with local adaptability, MoEMeta significantly advances few-shot relational learning. Extensive experiments and analyses on three KG benchmarks show that MoEMeta consistently outperforms existing baselines, achieving state-of-the-art performance.[1]

## 1 Introduction

Knowledge graphs (KGs) encode real-world knowledge in the form of factual triplets $(h, r, t)$, where each triplet represents a relationship $r$ between a head entity $h$ and a tail entity $t$ (Bollacker et al., 2008; Suchanek et al., 2007; Vrandečić and Krötzsch, 2014).KGs are crucial for powering various intelligent applications, such as question answering (Wu et al., 2016), Web search (Fang et al., 2017), and advanced recommendation systems (Cao et al., 2019). Despite their widespread adoption, real-world KGs are inherently incomplete. Although numerous KG completion methods (Bordes et al., 2013; Yang et al., 2015; Trouillon et al., 2016) have been proposed to infer missing facts based on observed triplets, their effectiveness is severely hindered by data sparsity, especially for long-tail or emerging relations with only a handful of known training triplets.

In response, *few-shot relational learning* (FSRL) has emerged as a promising direction for reasoning over few-shot relations with limited training triplets, enabling generalization to novel relations in data-scarce scenarios. Most existing FSRL methods (Chen et al., 2019; Jiang et al., 2021; Niu et al., 2021; Wu et al., 2023) are built upon model-agnostic meta-learning (MAML) (Finn et al., 2017), a gradient-based meta-learning framework that learns a set of global parameters as an initialization for rapid task adaptation using only one or a few steps of gradient descent updates. In FSRL, each task is relation-centric, comprising only a few training triplets of a single relation, and the objective is to

---

[*]Corresponding author.
[1]The code is available at: https://github.com/alexhw15/MoEMeta.

predict instances of novel relations not seen during meta-training. These methods focus on learning and encoding meta-knowledge among tasks to support rapid adaptation with minimal supervision.

**Limitations of prior work.** Despite promising results, current meta-learning based FSRL methods face two key limitations. First, they learn about meta-knowledge within individual tasks while neglecting shared patterns that exist across tasks. This limitation arises from the assumption that meta-training tasks are independent and identically distributed (i.i.d.), where a base model is adapted to each meta-training task using its support set, and then updated based on the performance on its query set. However, KG relations are heterogeneous and often semantically clustered, with some relations more closely related than others. For example, on Nell-One (Mitchell et al., 2018), `FatherOfPerson` and `BrotherOf` share a strong thematic overlap (family ties), whereas `ColorOf` pertains to a different type of attribute (physical properties). Ignoring such underlying commonalities among related tasks hinders model generalization in few-shot scenarios. While accounting for task variance has been explored in other few-shot domains like image classification (Li et al., 2019; Zhou et al., 2021), it remains largely unaddressed in the unique, non-i.i.d. context of relational learning.

Second, current methods are limited by their reliance on a single set of global parameters to encode meta-knowledge. With a shared initialization, a uniform adaptation process lacks the flexibility to capture the unique characteristics of individual tasks from only a few support triplets. This limitation is critical because relations and entities in KGs exhibit diverse interaction patterns (Wang et al., 2014; Ji et al., 2015), such as one-to-one, one-to-many, many-to-one, and many-to-many. An entity often reveals entirely different aspects depending on its relational context. For example, the entity `Elon Musk` participates in the `CeoOf` relation (a professional role) and the `FatherOfPerson` relation (a family role). Methods constrained by a shared initialization struggle to adapt to such locally distant contexts, and failing to modulate meta-knowledge for a specific task results in suboptimal adaptation.

**Present work and contributions.** To overcome these limitations, we propose **MoEMeta**, a novel meta-learning framework to tackle a critical, non-trivial challenge not addressed by prior FSRL work: *how to disentangle globally shared knowledge from task-specific contexts to achieve effective generalization and rapid adaptation under few-shot settings*. MoEMeta provides a principled solution through two novel components that work in synergy.

- **Global knowledge generalization**: To capture **what to generalize**, we introduce a **mixture-of-experts (MoE) model** to dynamically learn common relational patterns shared across tasks, referred to as **relational prototypes**, from a global pool of experts. The MoE is globally optimized, enabling the dynamic selection of experts that specialize in modeling the compositional nature of relations in KGs—an explicit and novel design for enhancing generalization.

- **Local context adaptation**: To address **what to adapt**, we propose a novel **task-tailored adaptation** mechanism that leverages task-specific structural contexts to fine-tune relation-meta and entity embeddings for local adaptation. Unlike general task-aware modulation methods (Vuorio et al., 2019; Abdollahzadeh et al., 2021), which rely on task mode prediction to modulate the meta-learned prior parameters, our adaptation operates at the embedding level, offering a lightweight yet effective way to capture diverse entity-relation interactions unique to individual tasks.

Together, these two innovations strike an effective balance between global knowledge generalization and fine-grained task-specific adaptation, which is crucial for FSRL with limited supervision. Experimental results demonstrate that MoEMeta achieves state-of-the-art performance on three KG benchmarks, significantly outperforming existing methods. In-depth ablation studies and visualizations further validate the efficacy of each component in MoEMeta.

## 2 Related Work

**Few-Shot Relation Learning.** Existing approaches to FSRL can be categorized into *metric-learning methods* and *meta-learning methods*. Metric-learning methods learn a similarity metric between the support and query sets for adaptation. Approaches in this category include GMatching (Xiong et al., 2018), designed for one-shot relation learning; FSKGC (Zhang et al., 2020), which extends to few-shot settings using a recurrent autoencoder to aggregate support instances; FAAN (Sheng et al., 2020), which employs a dynamic attention mechanism for improved one-hop neighbor aggregation; CSR (Huang et al., 2022), which utilizes subgraph matching for adaptation; and NP-FKGC (Luo et al., 2023), which leverages neural processes to address distribution shifts in unseen relations.

Most mainstream FSRL methods are based on meta-learning (Finn et al., 2017), which learns an initial meta-prior from existing tasks for rapid adaptation to new few-shot relations. MetaR (Chen et al., 2019) learns relation-specific meta-information by simply averaging support triplet representations. MetaP (Jiang et al., 2021) utilizes meta-pattern learning under one-shot settings. GANA (Niu et al., 2021) integrates meta-learning with TransH (Wang et al., 2014) and devises a gated, attentive neighbor aggregator to handle sparse neighborhood information. HiRe (Wu et al., 2023) exploits multi-granularity relational information to learn more generalizable meta-knowledge. RelAdapter (Ran et al., 2024) adopts a shallow feedforward network with a residual connection to adapt relation meta-knowledge within each target task. Despite their progress, these methods face two key limitations: (1) they often neglect common relational patterns when learning meta-knowledge during meta-training, and (2) they fail to effectively incorporate local, task-specific contexts essential for rapid adaptation. Our work addresses these critical gaps by proposing a principled meta-learning framework that disentangles globally shared knowledge from task-specific adaptation, enabling both effective generalization and fast adaptation in FSRL. For a more extensive review of related work, including KG embedding learning and general few-shot learning techniques, please refer to Appendix **??**.

**Mixture of Experts.** The mixture-of-experts (MoE) architecture, originally introduced by Jacobs et al. (1991), is a modular machine learning framework designed to improve performance by dynamically combining multiple specialized models, known as "experts". Sparse MoEs and their "soft" variants (Jordan and Jacobs, 1994; Puigcerver et al., 2024) have been widely adopted in deep learning, especially in large-scale models for natural language processing and computer vision, to improve model training efficiency and scalability through adaptive subnetwork selection or dynamic pruning (Shazeer et al., 2017; Du et al., 2022). In a related KG context, MoE has been applied to fuse different semantics of relations (structural ontology and textual descriptions) at the triplet level (Song et al., 2022). In contrast, our work innovatively proposes MoE as a meta-learner for FSRL that operates at the task level to learn specialized experts capturing diverse relational patterns, thereby enabling superior cross-task knowledge transfer and task adaptation.

## 3 Preliminaries

### 3.1 FSRL Problem Formulation

A knowledge graph (KG) can be represented as a collection of factual triplets $\mathcal{G} = \{(h, r, t) \in \mathcal{E} \times \mathcal{R} \times \mathcal{E}\}$, where $\mathcal{E}$ denotes the set of entities, and $\mathcal{R}$ denotes the set of relations. Each triplet $(h, r, t)$ indicates a relationship $r$ between the head entity $h$ and the tail entity $t$.

Given a novel relation $r \notin \mathcal{R}$ and its corresponding support set $\mathcal{S}_r = \{(h_i, r, t_i)\}_{i=1}^{K}$, where $h_i, t_i \in \mathcal{E}$, the objective of *few-shot relational learning* (FSRL) is to predict the missing tail entities in a query set $\mathcal{Q}_r = \{(h_j, r, ?)\}$, given a head entity $h_j \in \mathcal{E}$ and a relation $r$. For each query triplet $(h_j, r, ?) \in \mathcal{Q}_r$, a candidate set $\mathcal{C}_{h_j, r}$ is provided, and the goal is to rank the true tail entity highest among all candidates. The support set $\mathcal{S}_r$ and the query set $\mathcal{Q}_r$ jointly define a task $\mathcal{T}_r = (\mathcal{S}_r, \mathcal{Q}_r)$ for a relation $r$. In $K$-shot relation learning, the support set $\mathcal{S}_r$ contains only a few training triplets (typically $K = 1, 3, 5$).

### 3.2 Meta-Learning based FSRL

Given the strength of meta-learning in enabling rapid adaptation with minimal data, we build our approach upon a meta-learning based framework, involving two stages: meta-training and meta-testing. The model is expected to learn a meta-prior from seen tasks in the meta-training stage, which can then be adapted to novel tasks during the meta-testing stage.

**Meta-Training Stage.** During meta-training, an FSRL model is trained on a set of relations $\mathcal{R}_{\text{tr}}$ with the corresponding task data $\mathcal{D}_{\text{tr}} = \{\mathcal{T}_r \mid r \in \mathcal{R}_{\text{tr}}\}$. The model aims to learn a prior, which initializes a relation-meta learner for a new task. For each task $\mathcal{T}_r = (\mathcal{S}_r, \mathcal{Q}_r) \in \mathcal{D}_{\text{tr}}$, the learner extracts a task-specific relation-meta $\mathbf{R}_{\mathcal{T}_r} \in \mathbb{R}^d$ based on the support set $\mathcal{S}_r$, which is rapidly updated using a gradient meta $\mathbf{G}_{\mathcal{T}_r}$ computed from the loss on the support set:

$$\mathbf{R}_{\mathcal{T}_r} \leftarrow \mathbf{R}_{\mathcal{T}_r} - \alpha \mathbf{G}_{\mathcal{T}_r}, \tag{1}$$

where $\alpha$ is the learning rate. The updated relation-meta $\mathbf{R}_{\mathcal{T}_r}$ is then applied to the query set with the ground-truth tail entities, where the query loss is backpropagated to update both the prior and the entity embedding matrix.

**Meta-Testing Stage.** During meta-testing, the model is evaluated on novel relations $\mathcal{R}_{te}$ and their associated task data $\mathcal{D}_{te} = \{\mathcal{T}_r \mid r \in \mathcal{R}_{te}\}$, where $\mathcal{R}_{tr} \cap \mathcal{R}_{te} = \emptyset$. For each novel task $\mathcal{T}_r = (\mathcal{S}_r, \mathcal{Q}_r) \in \mathcal{D}_{te}$, the model uses the learned prior to initialize the relation-meta learner and compute an initial relation-meta $\mathbf{R}_{\mathcal{T}_r}$, which is then adapted using its support set $\mathcal{S}_r$ to obtain $\mathbf{R}'_{\mathcal{T}_r}$. Finally, for each query triplet $(h, r, ?)$, the model ranks all candidate tail entities $t$ in the candidate set $\mathcal{C}_{h,r}$ by computing a score $s(h, \mathbf{R}'_{\mathcal{T}_r}, t)$ for each candidate $t \in \mathcal{C}_{h,r}$ and sorting them in ascending order.

## 4 The Proposed MoEMeta Framework

This section presents our proposed **MoEMeta** framework. Its design is underpinned by two key insights: First, relations in real-world KGs exhibit heterogeneous characteristics, with some subsets of relations sharing greater similarity and stronger interconnections; Second, relations in KGs manifest diverse interaction patterns, such as one-to-many, many-to-one, and many-to-many. Building upon these insights, we introduce MoEMeta, a novel meta-learning framework that (1) utilizes an MoE model to dynamically learn relational prototypes and generate relation-meta for each task; and (2) incorporates a task-tailored adaptation mechanism to facilitate local adaptation to each task.

MoEMeta consists of three core components: (1) attentive neighbor aggregation, (2) MoE-based meta-knowledge learning, and (3) task-tailored local adaptation. Figure 1 illustrates the detailed process of calculating the loss on the support set, through these three components. In essence, MoEMeta($\boldsymbol{\Phi}, \boldsymbol{\eta}$) contains two sets of learnable parameters: the parameters in attentive neighbor aggregation and MoE-based meta-knowledge learning are globally optimized during meta-training, denoted as $\boldsymbol{\Phi}$; while the parameters in task-tailored local adaptation are locally optimized within each task, denoted as $\boldsymbol{\eta}$.

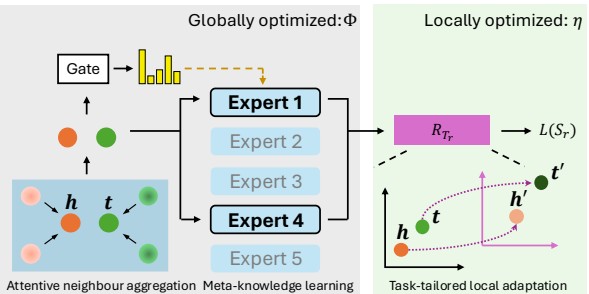

Figure 1: The computation of the support loss. The support loss $\mathcal{L}(\mathcal{S}_r)$ is used to update relation-meta $\mathbf{R}_{\mathcal{T}_r}$, which is applied to the query set for model optimization.

### 4.1 Attentive Neighbor Aggregation

Given a $K$-shot task $\mathcal{T}_r$ with its support set $\mathcal{S}_r = \{(h_i, r, t_i)\}_{i=1}^{K}$, we propose an attentive neighbor encoder that aggregates the neighborhood information for both head entity $h_i$ and tail entity $t_i$. Specifically, for each entity, the encoder processes its neighboring tuples (neighboring entities and relations), combined with its own embeddings, to generate an enriched representation.

For a given entity $e$, its one-hop neighbors are represented as a set of neighboring tuples $\{(r_i, e'_i)\}_{i=1}^{n}$, where $n$ is the total number of neighbors, $e'_i$ is the neighboring entity and $r_i$ denotes the relation between $e'_i$ and $e$. We first concatenate the embeddings of $r_i$ and $e'_i$ to form the embedding of the $i$-th tuple, $\mathbf{c}_i = \mathbf{r}_i \oplus \mathbf{e}'_i$, where $\oplus$ denotes concatenation of two vectors. To capture interactions among neighbors, we then apply a linear transformation followed by a non-linear activation function:

$$\mathbf{c}'_i = \text{ReLU}(\mathbf{W}\mathbf{c}_i), \tag{2}$$

where $\mathbf{W}$ is a learnable weight matrix. To enable the encoder to attend to informative neighbors, we compute an attention-like gate value $g_i$ for each neighboring tuple using a sigmoid function:

$$g_i = \sigma(\boldsymbol{\beta}^T \mathbf{c}'_i), \tag{3}$$

where $\boldsymbol{\beta}$ is a learnable parameter vector. The gate values control the influence of different neighbors in the aggregation process. Finally, the weighted neighbor embeddings are aggregated and combined with the entity's own embedding $\mathbf{e}$ to obtain the final embedding:

$$\mathbf{e} = \text{Aggregate}(\{g_i \cdot \mathbf{c}'_i\}_{i=1}^{n}) + \mathbf{e}. \tag{4}$$

The neighbor encoder is applied to head entity $h$ and tail entity $t$ independently to ensure that the final embedding reflects both the entity's intrinsic information and its local neighborhood structure.

## 4.2 MoE-based Meta-Knowledge Learning

To overcome the limitation of existing methods that neglect shared meta-knowledge across related tasks, we introduce a globally learned MoE model that captures common relational patterns, termed *relational prototypes*. For each task, a subset of relevant experts is dynamically selected from a shared pool based on its support triplets to learn the relation-meta. This mechanism enables adaptive relation decomposition and expert specialization, facilitating the learning of diverse, compositional relational patterns and promoting effective generalization across tasks.

Specifically, we design a sparsely-gated MoE model consisting of $M$ experts. Each expert within the MoE is architecturally identical to a standard MLP (multilayer perceptron). For a given support triplet, a gating mechanism dynamically routes the triplet to a selected subset of experts, assigning weights to each expert in the subset based on its relevance. The outputs of the top-$N$ experts are then aggregated according to their weights, resulting in a representation of the relation associated with the triplet. To compute the relation-meta for a support set, we average the relation representations derived from all triplets within the support set. This approach enables the MoE to retrieve the most relevant meta-knowledge from the shared expert pool to model diverse relational patterns.

Given a $K$-shot task $\mathcal{T}_r$ and its support set $\mathcal{S}_r = \{(h_i, r, t_i)\}_{i=1}^{K}$, the head and tail entities of each triplet, $h_i$ and $t_i$, are augmented via our attentive neighbor aggregator to obtain the embeddings $\mathbf{h}_i$ and $\mathbf{t}_i$. The relation representation $\mathbf{r}_i$ derived from $(h_i, t_i)$ can be computed by our MoE as follows:

$$\mathbf{r}_i = \frac{1}{N} \sum_{j=1}^{M} g_{i,j} f_j(\mathbf{h}_i, \mathbf{t}_i; \boldsymbol{\theta}_j), \tag{5}$$

$$g_{i,j} = \begin{cases} s_{i,j}, & \text{if } s_{i,j} \in \text{Top}_i\big(\{s_{i,j} \mid 1 \leq j \leq M\}, N\big), \\ 0, & \text{otherwise,} \end{cases} \tag{6}$$

$$s_{i,j} = \text{softmax}(\text{Gate}(\mathbf{h}_i, \mathbf{t}_i; \boldsymbol{\theta}_g)), \tag{7}$$

where each $f_j(\cdot; \boldsymbol{\theta}_j)$ is an expert network and $M$ denotes the total number of experts. For each support triplet $i$, a gating network $\text{Gate}(\cdot; \boldsymbol{\theta}_g)$ computes a relevance score $s_{i,j}$ for each expert $j$, conditioned on the corresponding inputs $\mathbf{h}_i$ and $\mathbf{t}_i$. These scores are then normalized via a softmax function over all $M$ experts. The function $\text{Top}_i(\cdot, N)$ selects only the top $N$ highest scores among $M$ experts for each triplet $i$, setting the remaining scores to zero. This sparsity mechanism enables the model to selectively activate only a subset of experts for each support triplet, facilitating the learning of more specialized relational prototypes globally shared across tasks.

The final relation-meta is computed as the average of relation representations from all support triplets:

$$\mathbf{R}_{\mathcal{T}_r} = \frac{1}{K} \sum_{i=1}^{K} \mathbf{r}_i \tag{8}$$

## 4.3 Task-Tailored Local Adaptation

As discussed earlier, the global parameters alone—$\mathbf{W}$ and $\boldsymbol{\beta}$ in attentive neighbor aggregation, and $\{\boldsymbol{\theta}_i\}_{i=1}^{M}$ and $\boldsymbol{\theta}_g$ in meta-knowledge learning—may be insufficient for effective task-specific adaptation from only a few support triplets. The unique characteristics of individual tasks are difficult to capture with a shared initialization. To address this issue, we propose a dynamic learnable projection mechanism that facilitates local adaptation, allowing task-specific relational learning tailored to each task.

Specifically, for each task, MoEMeta maintains a task-specific set of local parameters, denoted as $\boldsymbol{\eta}$, namely three randomly initialized projection vectors, $\mathbf{p}_h$, $\mathbf{p}_r$, and $\mathbf{p}_t$, which project the embeddings $\mathbf{h}_i$, $\mathbf{R}_{\mathcal{T}_r}$, and $\mathbf{t}_i$ into a task-specific subspace. Inspired by TransD (Ji et al., 2015), which addresses the limitations of fixed mapping strategies using per-entity and per-relation projections, our approach leverages task-specific projections to avoid overparametrization under few-shot settings. As all triplets within a task share the same relation, we adopt a unified projection vector $\mathbf{p}_h$ for all head entities, and $\mathbf{p}_t$ for all tail entities within the task. This provides a lightweight solution for enabling efficient task-specific adaptation with minimal parameter overhead.

Unlike previous FSRL methods including RelAdapter (Ran et al., 2024), which do not account for diverse interactions between entities and relations tailored to different tasks, our approach allows for a relation-specific adaptation, with projection operations defined below:

$$\mathbf{h}'_i = \mathbf{h}_i + (\mathbf{p}_h^\top \mathbf{R}_{\mathcal{T}_r}) \cdot \mathbf{R}_{\mathcal{T}_r}, \tag{9}$$

$$\mathbf{R}'_{\mathcal{T}_r} = \mathbf{R}_{\mathcal{T}_r} + (\mathbf{p}_r^\top \mathbf{R}_{\mathcal{T}_r}) \cdot \mathbf{R}_{\mathcal{T}_r}, \tag{10}$$

$$\mathbf{t}'_i = \mathbf{t}_i + (\mathbf{p}_t^\top \mathbf{R}_{\mathcal{T}_r}) \cdot \mathbf{R}_{\mathcal{T}_r}. \tag{11}$$

Above, the dot product $\mathbf{p}_x^\top \mathbf{R}_{\mathcal{T}_r}$ yields a scalar alignment score between each projection vector $\mathbf{p}_x, x \in \{h, r, t\}$, and the relation-meta $\mathbf{R}_{\mathcal{T}_r}$. This scalar score is then used to scale $\mathbf{R}_{\mathcal{T}_r}$, yielding an offset vector that modulates corresponding features in the entity embeddings and relation-meta. The updated embeddings ($\mathbf{h}'_i$, $\mathbf{R}'_{\mathcal{T}_r}$, $\mathbf{t}'_i$) thus capture task-specific nuances while maintaining consistency with the generalizable meta-knowledge. Although the three projection vectors could functionally be merged into a single perturbation term, maintaining separate projections for the head entity, relation-meta, and tail entity enhances modularity and flexibility. This design allows for more expressive or heterogeneous projection functions that can incorporate richer inductive biases or task-specific modeling needs in future extensions. Overall, this local adaptation scheme ensures that each task can adaptively focus on its unique aspects while preserving shared commonalities across related tasks.

To further optimize the adapted embeddings $\mathbf{h}'_i$, $\mathbf{R}'_{\mathcal{T}_r}$, and $\mathbf{t}'_i$, we define a scoring function in the projected space to measure the plausibility of each triplet $(h_i, r, t_i)$, which is given by:

$$\text{score}(h_i, t_i) = \left\| \mathbf{h}'_i + \mathbf{R}'_{\mathcal{T}_r} - \mathbf{t}'_i \right\|_2, \tag{12}$$

where $\|\cdot\|_2$ denotes the $\ell_2$ norm. $\mathbf{h}'_i$, $\mathbf{R}'_{\mathcal{T}_r}$, and $\mathbf{t}'_i$ are the projected embeddings of the head entity, relation-meta, and tail entity, respectively. This formulation ensures that, in the task-specific projection space, the distance between the adapted head entity embedding $\mathbf{h}'_i$ and the adapted tail entity embedding $\mathbf{t}'_i$ is minimized after being translated by the relation-meta embedding $\mathbf{R}'_{\mathcal{T}_r}$.

Consider the entire support set, we define a margin-based loss function as follows:

$$\mathcal{L}(\mathcal{S}_r) = \sum_{(h_i, r, t_i) \in \mathcal{S}_r} \max\{0, \text{score}(h_i, t_i) + \gamma - \text{score}(h_i, t'_i)\}, \tag{13}$$

where $\gamma$ is a margin hyperparameter, and $(h_i, r, t'_i)$ represents a negative triplet generated by replacing the tail entity $t_i$ with a randomly sampled entity $t'_i \in \mathcal{E}$ such that $(h_i, r, t'_i) \notin \mathcal{G}$.

Once the gradients of the support loss $\mathcal{L}(\mathcal{S}_r)$ are computed, we update the projection vectors $\mathbf{p}_h$, $\mathbf{p}_r$, $\mathbf{p}_t$, together with the relation-meta $\mathbf{R}_{\mathcal{T}_r}$, to ensure generalization to the support set. Specifically, the projection vectors can be updated in a unified manner as follows:

$$\mathbf{p}_x \leftarrow \mathbf{p}_x - \alpha \frac{\partial \mathcal{L}(\mathcal{S}_r)}{\partial \mathbf{p}_x}, \quad x \in \{h, r, t\}. \tag{14}$$

Similarly, the relation-meta $\mathbf{R}_{\mathcal{T}_r}$ is updated as:

$$\mathbf{R}_{\mathcal{T}_r} \leftarrow \mathbf{R}_{\mathcal{T}_r} - \alpha \frac{\partial \mathcal{L}(\mathcal{S}_r)}{\partial \mathbf{R}_{\mathcal{T}_r}}, \tag{15}$$

where $\alpha$ is the learning rate. These updates adapt the model using the support set, which is then evaluated on the query set $\mathcal{Q}_r$. For each query triplet $(h_q, r, t_q) \in \mathcal{Q}_r$, we evaluate its plausibility for each $t_q$ from the candidate set $C_{h_q, r}$, using the score function Eq. (12). Then, the query loss is similarly defined as:

$$\mathcal{L}(\mathcal{Q}_r) = \sum_{(h_q, r, t_q) \in \mathcal{Q}_r} \max\{0, \text{score}(h_q, t_q) + \gamma - \text{score}(h_q, t'_q)\}, \tag{16}$$

where $(h_q, r, t'_q)$ is a negative triplet generated by corrupting the tail entity $t_q$ with a randomly sampled entity $t'_q \in \mathcal{E}$. The optimization objective for training the entire model is to minimize $\mathcal{L}(\mathcal{Q}_r)$.

The task-specific projection vectors $\mathbf{p}_h$, $\mathbf{p}_r$, and $\mathbf{p}_t$ are randomly initialized and trained for each task on the support set $\mathcal{S}_r$, enabling effective local adaptation. This mechanism provides two key advantages. (1) Task-specific adaptation: By projecting embeddings into a task-specific latent space, the model captures fine-grained task-specific local contexts. (2) Complementarity to MoE: While the MoE captures globally shared relational prototypes, dynamic learnable projections provide finer-grained local adaptation, thereby improving model generalization under few-shot settings.

## 4.4 MoEMeta Meta-Training and Meta-Testing

During meta-training, MoEMeta optimizes two sets of parameters: global parameters $\Phi = \{\mathbf{W}, \boldsymbol{\beta}, \{\boldsymbol{\theta}\}_{i=1}^M, \boldsymbol{\theta}_g\}$, for attentive neighbor aggregation and MoE-based meta-knowledge learning, which are globally optimized; and task-specific adaptation parameters $\boldsymbol{\eta} = \{\mathbf{p}_h, \mathbf{p}_r, \mathbf{p}_t\}$, which are optimized locally for each task. The meta-training algorithm of MoEMeta is detailed in Algorithm 1.

During meta-testing, the learned global parameters $\Phi$ are used to initialize the model and remain frozen. For each novel task, only the local parameters $\boldsymbol{\eta}$ are fine-tuned on the support set before evaluation on the query set by ranking candidate tail entities, following the same inference procedure as in meta-training. The complexity analysis of MoEMeta's components is provided in Appendix A.

---

**Algorithm 1:** Meta-training Algorithm of MoEMeta

---

**Input:** Meta-training task set $\mathcal{D}_{\text{tr}}$, knowledge graph $\mathcal{G}$
**Output:** Optimized global parameters $\Phi$

1 Initialize global parameters $\Phi = \{\mathbf{W}, \boldsymbol{\beta}, \{\boldsymbol{\theta}\}_{i=1}^M, \boldsymbol{\theta}_g\}$;
2 **while** *not converged* **do**
3      Sample a task $\mathcal{T}_r = \{\mathcal{S}_r, \mathcal{Q}_r\} \in \mathcal{D}_{\text{tr}}$;
4      **foreach** $(h_i, r, t_i) \in \mathcal{S}_r$ **do**
         `// Attentive Neighbor Aggregation`
5          Attentively aggregate neighbors using Eq. 2–Eq. 4;
         `// MoE-based Meta-Knowledge Learning`
6          Compute expert scores via Eq. 7;
7          Select top-$N$ experts and compute relation representation $\mathbf{r}_i$;
8      Compute the initial relation-meta $\mathbf{R}_{\mathcal{T}_r}$ via Eq. 8;
     `// Inner Loop: Task-Tailored Local Adaptation`
9      Initialize task-specific adaptation parameters $\boldsymbol{\eta} = \{\mathbf{p}_h, \mathbf{p}_r, \mathbf{p}_t\}$;
10     **foreach** $(h_i, r, t_i) \in \mathcal{S}_r$ **do**
11         Compute projected embeddings using Eq. 9–Eq. 11;
12     Compute the support loss $\mathcal{L}(\mathcal{S}_r)$ via Eq. 13;
13     Update task-specific projection vectors $\mathbf{p}_h, \mathbf{p}_r, \mathbf{p}_t$ via Eq. 14;
14     Update relation-meta $\mathbf{R}_{\mathcal{T}_r}$ via Eq. 15;
     `// Outer Loop: Meta-Update`
15     Compute the query loss $\mathcal{L}(\mathcal{Q}_r)$ over all $(h_q, r, t_q) \in \mathcal{Q}_r$ via Eq. 16;
16     Update global parameters $\Phi$ via gradient descent on $\mathcal{L}(\mathcal{Q}_r)$.

---

## 5 Experiments

**Datasets.** We evaluate our method on three widely used KG benchmarks specifically for few-shot relational learning: Nell-One and Wiki-One (Xiong et al., 2018), as well as FB15K-One (Ran et al., 2024). The training/validation/test splits include 51/5/11, 133/16/34, and 75/11/33 tasks on Nell-One, Wiki-One, and FB15K-One, respectively. Detailed dataset statistics are summarized in Table 1.

**Evaluation Metrics.** We report both MRR (mean reciprocal rank) and Hits@$k$ ($k = 1, 5, 10$). MRR captures the average reciprocal rank of correct entities, while Hits@$k$ measures the proportion of correct entities appearing in the top $k$ predictions. Following prior studies, our method is evaluated against baselines under the 1-shot and 5-shot settings

Table 1: Statistics of datasets.

| Dataset | #Relations | #Entities | #Triplets | #Tasks |
|---------|-----------|-----------|-----------|--------|
| Nell-One | 358 | 68,545 | 181,109 | 67 |
| Wiki-One | 822 | 4,838,244 | 5,859,240 | 183 |
| FB15K-One | 237 | 14,541 | 281,624 | 119 |

on Nell-One and Wiki-One, which are commonly used for evaluating FSRL, and under the 3-shot setting on FB15K-One, following the setup of RelAdapter (Ran et al., 2024).

**Baselines.** We compare our proposed method with two categories of state-of-the-art baselines:

- **Conventional KG embedding methods**: This group focuses on modeling relational structures in KGs to learn entity and relation embeddings. We compare with four widely-used baselines including TransE (Bordes et al., 2013), TransD (Ji et al., 2015), DistMult (Yang et al., 2015), and

Table 2: Comparison against state-of-the-art methods on Nell-One and Wiki-One. MetaR-I and MetaR-P indicate the In-train and Pre-train of MetaR (Chen et al., 2019).

| Methods | Nell-One | | | | | | | | Wiki-One | | | | | | | |
|---|---|---|---|---|---|---|---|---|---|---|---|---|---|---|---|---|
| | MRR | | Hits@1 | | Hits@5 | | Hits@10 | | MRR | | Hits@1 | | Hits@5 | | Hits@10 | |
| | 1-shot | 5-shot | 1-shot | 5-shot | 1-shot | 5-shot | 1-shot | 5-shot | 1-shot | 5-shot | 1-shot | 5-shot | 1-shot | 5-shot | 1-shot | 5-shot |
| TransE | 0.105 | 0.168 | 0.041 | 0.082 | 0.111 | 0.186 | 0.226 | 0.345 | 0.036 | 0.052 | 0.011 | 0.042 | 0.024 | 0.057 | 0.059 | 0.090 |
| TransD | 0.159 | 0.276 | 0.125 | 0.159 | 0.162 | 0.322 | 0.234 | 0.431 | 0.063 | 0.101 | 0.034 | 0.061 | 0.062 | 0.104 | 0.135 | 0.181 |
| DistMult | 0.165 | 0.214 | 0.106 | 0.140 | 0.174 | 0.246 | 0.285 | 0.319 | 0.046 | 0.077 | 0.014 | 0.035 | 0.034 | 0.078 | 0.087 | 0.134 |
| ComplEx | 0.179 | 0.239 | 0.112 | 0.176 | 0.212 | 0.253 | 0.299 | 0.364 | 0.055 | 0.070 | 0.021 | 0.030 | 0.044 | 0.063 | 0.100 | 0.124 |
| GMatching | 0.185 | 0.201 | 0.119 | 0.143 | 0.260 | 0.264 | 0.313 | 0.311 | 0.200 | - | 0.120 | - | 0.272 | - | 0.336 | - |
| FSKGC | - | 0.184 | - | 0.136 | - | 0.234 | - | 0.272 | - | 0.158 | - | 0.097 | - | 0.206 | - | 0.287 |
| FAAN | - | 0.279 | - | 0.200 | - | 0.364 | - | 0.428 | - | 0.341 | - | 0.281 | - | 0.395 | - | 0.436 |
| MetaR-I | 0.250 | 0.261 | 0.170 | 0.168 | 0.336 | 0.350 | 0.401 | 0.437 | 0.193 | 0.221 | 0.152 | 0.178 | 0.233 | 0.264 | 0.280 | 0.302 |
| MetaR-P | 0.164 | 0.209 | 0.093 | 0.141 | 0.238 | 0.280 | 0.331 | 0.355 | 0.314 | 0.323 | 0.266 | 0.270 | 0.375 | 0.385 | 0.404 | 0.418 |
| MetaP† | 0.232 | - | 0.179 | - | 0.281 | - | 0.330 | - | - | - | - | - | - | - | - | - |
| GANA | 0.236 | 0.245 | 0.173 | 0.166 | 0.293 | 0.334 | 0.347 | 0.390 | 0.260 | 0.261 | 0.221 | 0.317 | 0.307 | 0.333 | 0.334 | 0.384 |
| HiRe | 0.288 | 0.306 | 0.184 | 0.207 | 0.403 | 0.439 | 0.472 | 0.520 | 0.322 | 0.371 | 0.271 | 0.319 | 0.383 | 0.419 | 0.433 | 0.469 |
| RelAdapter† | - | - | - | - | - | - | - | - | 0.247 | 0.305 | 0.209 | 0.245 | 0.281 | 0.365 | 0.307 | 0.415 |
| MoEMeta | 0.322• | 0.339• | 0.228• | 0.236• | 0.429• | 0.440 | 0.501• | 0.523 | 0.325• | 0.377• | 0.276 | 0.324 | 0.385 | 0.420 | 0.434 | 0.473 |

†: MetaP and RelAdapter's results are not fully available due to unreleased pre-trained information.

ComplEx (Trouillon et al., 2016). Using OpenKE (Han et al., 2018), these models are reproduced with hyperparameters specified in the original papers.

- **Few-shot relational learning methods**: We compare our method against state-of-the-art FSRL baselines evaluated under the same framework, including (1) metric-learning methods: GMatching (Xiong et al., 2018), FAAN (Sheng et al., 2020), FSKGC (Zhang et al., 2020); and (2) strong meta-learning competitors: MetaR (Chen et al., 2019), GANA (Niu et al., 2021), HiRe (Wu et al., 2023), and RelAdapter (Ran et al., 2024). We exclude methods like CSR (Huang et al., 2022) and NP-FKGC (Luo et al., 2023) as they employ a different training and evaluation protocol.

All results are obtained after models are trained using all triplets from background relations (Xiong et al., 2018) and meta-training tasks, and then evaluated on meta-testing tasks.

**Experimental Setup.** Following prior work, we set the embedding dimension to 100 for Nell-One and FB15K-One, and 50 for Wiki-One, initialized using TransE-pretrained weights provided by GMatching (Xiong et al., 2018). The number of neighbors per entity is capped at 50 for aggregation. For MoE, each expert network is a two-layer MLP (hidden dimension: 64, ReLU). The gating network is also a two-layer MLP (hidden dimension: 64, output dimension: 1, and ReLU). We set the number of experts $M$ as 32 and the number of selected experts $N$ as 5. The margin $\gamma$ in Eq. 16 is set to 1. MoEMeta is trained using the Adam optimizer (batch size: $1,024$, learning rate: $0.001$). All results are averaged over 5 independent runs with different random seeds. All models are implemented in PyTorch and trained on a single Tesla P100 GPU. More details for reproducing our results are provided in Appendix B.1.

## 5.1 Comparison with Baseline Methods

Tables 2 and 3 compare MoEMeta against state-of-the-art methods on Nell-One and Wiki-One under the 1-shot and 5-shot settings, and on FB15K-One under the 3-shot setting. The best and the second best performers are respectively marked by grey and underline. Our proposed MoEMeta consistently achieves state-of-the-art results across all datasets and settings. On Nell-One, MoEMeta surpasses HiRe, the second best performer, by 11.8% MRR, 23.9% Hits@1, 6.4% Hits@5, and 6.1% Hits@10 under 1-shot setting, and 10.8% MRR, 14.0% Hits@1, 2.3% Hits@5, and 0.6% Hits@10 under 5-shot setting. On FB15K-One, MoEMeta outperforms the second best method by 4.4% MRR, 0.3% Hits@1, and 13.2% Hits@5 under 3-shot setting. We further conduct paired t-tests between MoEMeta and the second best performers on all datasets. The improvements of MoEMeta over the second best method with statistical significance at the 0.05 level are marked with •. These results highlight MoEMeta's effectiveness in learning

Table 3: Comparison under 3-shot setting on FB15K-One.

| Methods | FB15K-One | | |
|---|---|---|---|
| | MRR | Hits@1 | Hits@10 |
| TransE | 0.294 | 0.204 | 0.437 |
| TransD | 0.301 | 0.203 | 0.439 |
| DistMult | 0.234 | 0.208 | 0.364 |
| ComplEx | 0.239 | 0.205 | 0.359 |
| GMatching | 0.309 | 0.245 | 0.441 |
| MetaR | 0.368 | 0.251 | 0.536 |
| FAAN | 0.363 | 0.279 | 0.542 |
| GANA | 0.388 | 0.301 | 0.553 |
| HiRe | 0.378 | 0.281 | 0.571 |
| RelAdapter | 0.405 | 0.297 | 0.575 |
| MoEMeta | 0.423 • | 0.302 | 0.651 • |

Table 4: Ablation study on Nell-One and FB15K-One.

| Ablation on | Nell-One | | | | | | | | FB15K-One | | |
|---|---|---|---|---|---|---|---|---|---|---|---|
| | 1-shot | | | | 5-shot | | | | 3-shot | | |
| | MRR | Hits@1 | Hits@5 | Hits@10 | MRR | Hits@1 | Hits@5 | Hits@10 | MRR | Hits@1 | Hits@10 |
| MoEMeta | 0.322 | 0.228 | 0.429 | 0.501 | 0.339 | 0.236 | 0.440 | 0.523 | 0.422 | 0.298 | 0.652 |
| w/o N.A | 0.311 | 0.219 | 0.418 | 0.489 | 0.328 | 0.231 | 0.436 | 0.509 | 0.413 | 0.286 | 0.631 |
| w/o MoE | 0.291 | 0.199 | 0.392 | 0.467 | 0.293 | 0.201 | 0.399 | 0.484 | 0.401 | 0.273 | 0.624 |
| w/o L.A | 0.301 | 0.206 | 0.398 | 0.476 | 0.315 | 0.213 | 0.415 | 0.501 | 0.408 | 0.285 | 0.628 |

N.A indicates "attentive neighbor aggregation" and L.A indicates "task-tailored local adaptation".

generalizable meta-knowledge that adapts well to novel relations and proving its competitiveness against other baselines.

Across three datasets, the most substantial improvements are on Nell-One, particularly in MRR under both 1-shot and 5-shot settings. The primary gain is attributed to the increase in the challenging Hits@1 metric, which gauges the model's ability to rank the correct tail entity as the top candidate among all choices. This is especially notable since triplets in Nell-One are constructed in natural language, providing richer semantics that can be more effectively captured by our relational prototypes. These results validate the efficacy of our proposed MoE-based relation-meta learner in learning globally shared, fine-grained relational patterns and adapting these prototypes to new, unseen tasks. We additionally provide a runtime analysis for both meta-training and meta-testing in Appendix B.2.

## 5.2 Ablation Studies

To assess the contribution of MoEMeta's three key components, attentive neighbor aggregation, MoE-based meta-knowledge learning, and task-tailored local adaptation, we report ablation results on Nell-One under the 1-shot and 5-shot settings, and FB15K-One under the 3-shot setting in Table 4.

- **Removing attentive Neighbor Aggregation (w/o N.A)**, which removes the first term in Eq. 4, results in a moderate performance decrease across all settings. This suggests the importance of incorporating neighboring information to enhance entity representations.
- **Replacing MoE (w/o MoE)** with an MLP-based relation-meta learner from (Chen et al., 2019) leads to a significant performance decline; MRR drops by 9.6% and 13.6% under 1-shot and 5-shot settings on Nell-One, and 5.0% under 3-shot setting on FB15K-One. This underscores MoE's indispensable role in acquiring relational prototypes as generalizable meta-knowledge.
- **Ablating task-tailored Local Adaptation (w/o L.A)**, where the learned relation-meta is directly applied to the query set, also causes a significant performance drop across datasets. This suggests that global meta-knowledge alone is insufficient, highlighting the crucial role of task-tailored adaptation for capturing local distinct patterns and improving generalization across tasks.

## 5.3 Analysis on Complex Relations

To further evaluate the significance of task-tailored local adaptation for complex relations, we conduct a detailed analysis on Nell-One under the 5-shot setting by categorizing the meta-testing set into three relation types: 1-N, N-1, and N-N. In particular, we compare MoEMeta with a variant that removes task-tailored local adaptation (w/o L.A). As shown in Table 5, the removal of local adaptation leads to a substantially larger performance drop on 1-N and N-N relations than on N-1 relations. This is because, in N-1 relations, the tail entity is relatively predictable, with a smaller search space and more stable mapping patterns. In contrast, 1-N and N-N relations exhibit higher ambiguity and a more dispersed distribution of tail entities, making them more difficult to predict. Notably, removing task-tailored adaptation results in the largest performance drop for the N-N relation type. This analysis confirms the critical role of our task-tailored adaptation mechanism in enhancing generalization to complex and highly ambiguous relational patterns.

In addition, we visualize the distributions of gating values of different relations in Figure 2 to provide a more intuitive understanding of why learning relational prototypes works. As observed, relations with similar patterns (in the upper figure) exhibit a comparable gating value distribution and activate similar subsets of experts, whereas dissimilar relations (in the lower figure) demonstrate a markedly different pattern. This observation highlights our model's ability to distinguish diverse relational patterns but capture globally shared relational prototypes across related tasks.

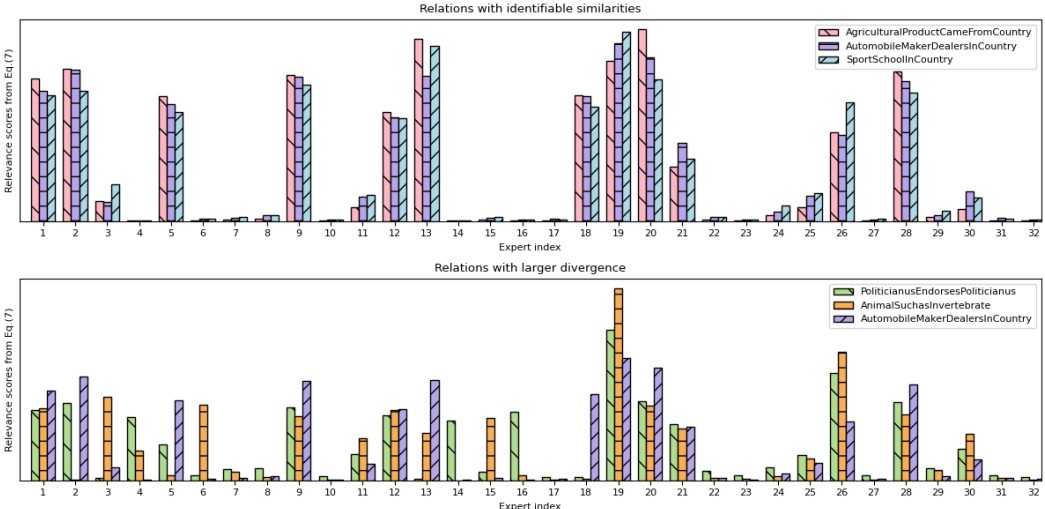

Figure 2: Two gating value distributions over 32 experts generated by our MoE-based meta-knowledge learner. The upper figure shows that three similar relations exhibit similar gating value distributions, suggesting that a similar set of experts are activated. In contrast, the lower figure demonstrates that for relations with distinct relational patterns, the activated experts differ significantly. This highlights the presence of shared relational patterns, underscoring the necessity of learning relation prototypes.

Table 5: Performance analysis under 5-shot setting on Nell-One for 1-N, N-1 and N-N relation types.

| Methods | Rel. types | MRR | Hits@1 | Hits@5 | Hits@10 |
|---------|-----------|-----|--------|--------|---------|
| MoEMeta | 1-N | 0.406 | 0.310 | 0.506 | 0.577 |
|         | N-1 | 0.945 | 0.913 | 0.986 | 0.986 |
|         | N-N | 0.404 | 0.279 | 0.536 | 0.618 |
| w/o L.A | 1-N | 0.391 | 0.292 | 0.500 | 0.565 |
|         | N-1 | 0.874 | 0.871 | 0.986 | 0.986 |
|         | N-N | 0.372 | 0.256 | 0.485 | 0.586 |

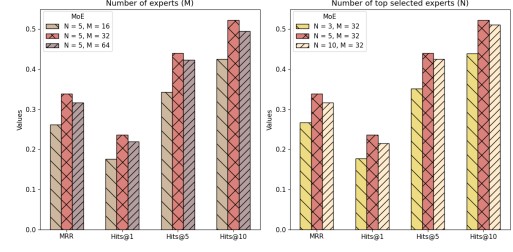

Figure 3: Hyperparameter sensitivity study of MoE under 5-shot setting on Nell-One.

## 5.4 Hyperparameter Analysis on MoE

To assess the impact of two MoE hyperparameters, the number of experts $M$ and the number of top selected experts $N$, we conduct a sensitivity analysis on Nell-One under the 5-shot setting as a case study. When fixing $N = 5$ and varying $M \in \{16, 32, 64\}$, MoEMeta achieves the best performance with 32 experts. This aligns with the guideline of setting the number of experts $M$ smaller than the total number of tasks to encourage learning shared relational prototypes and prevent overfitting with limited support triplets. When fixing $M = 32$ and varying $N \in \{3, 5, 10\}$, we observe that selecting slightly more experts (e.g., 5 vs. 3) leads to improved performance, yet increasing $N$ to 10 causes only a slight drop, suggesting that MoEMeta is relatively insensitive to $N$ within a reasonable range.

## 6 Conclusion

This work presents MoEMeta, a novel meta-learning framework for few-shot relational learning in KGs. By integrating an MoE model to capture globally shared relational prototypes with a task-tailored adaptation mechanism to refine embeddings locally, MoEMeta effectively balances cross-task generalization and rapid task-specific adaptation. Extensive experiments on three real-world KG datasets demonstrate that MoEMeta consistently outperforms state-of-the-art methods, showcasing its effectiveness for few-shot relation learning. Comprehensive ablation studies and visualizations further validate the importance of MoEMeta's key components, confirming its efficacy in modeling complex relational data in KGs under limited supervision.

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

# Appendix

## A Complexity Analysis

In this section, we analyze the computational complexity of our proposed MoEMeta for its key components. Table 6 summarizes the computational complexity of each component, where $d$ denotes the dimension of entity embeddings, $n$ denotes the number of neighbors in attentive neighbor aggregation, and $K$ denotes the number of support triplets per task. For MoE-based meta-knowledge learning, $M$ indicates the total number of experts, and $N$ indicates the number of experts activated in each forward pass.

As shown in Table 6, the complexity of MoEMeta scales linearly with the number of neighbors per entity during the aggregation stage ($\mathcal{O}(Knd^2)$) and linearly with the number of support triplets during MoE-based meta-knowledge learning ($\mathcal{O}(NKd^2)$). The task-tailored local adaptation introduces only a minimal overhead of ($\mathcal{O}(d)$) in both parameters and computation. Overall, the computational cost is dominated by neighbor aggregation ($\mathcal{O}(Knd^2)$), which is an essential component in current state-of-the-art FSRL methods. The additional computational and parameter overhead introduced by MoEMeta is negligible, making it efficient for FSRL in KGs.

Table 6: Complexity analysis of MoEMeta components with respect to the number of model parameters and the number of multiplication operations in each epoch. $d$ denotes the dimension of entity embeddings. $n$ denotes the number of neighbors per entity, and $K$ denotes the number of support triplets per task, where $K \ll n, d$. $M$ is the total number of experts, while $N$ is the number of experts activated in each forward pass.

| Component | # Parameters | # Multiplication Operations |
|---|---|---|
| Attentive neighbor aggregation | $\mathcal{O}(d^2)$ | $\mathcal{O}(Knd^2)$ |
| MoE-based meta-knowledge learning | $\mathcal{O}(Md^2)$ | $\mathcal{O}(NKd^2)$ |
| Task-tailored local adaptation | $\mathcal{O}(d)$ | $\mathcal{O}(d)$ |
| Total | $\mathcal{O}(Md^2)$ | $\mathcal{O}(nKd^2)$ |

## B Additional Experimental Details and Results

### B.1 Reproducibility Details

In our experiments, our method and all baselines are compared using the same evaluation protocol. The results of baseline methods are reproduced by using their official open-sourced implementations or adopted from their respective papers.

For MetaR (Chen et al., 2019) (both In-Train and Pre-Train)[2],FAAN (Sheng et al., 2020)[3], MetaP (Jiang et al., 2021)[4] and HiRe (Wu et al., 2023)[5], we directly use the results reported in their papers. The results for GMatching (Xiong et al., 2018)[6] under 1-shot and 5-shot settings are taken from (Chen et al., 2019). Since FSKGC (Zhang et al., 2020)[7] was originally evaluated under different conditions (with a smaller candidate set), we use the re-implemented results provided by FAAN to ensure consistency. For GANA (Niu et al., 2021)[8], we reproduce its results by setting the number of neighbors to 50, aligning with the settings of other methods. We reproduce the results of RelAdapter (Ran et al., 2024)[9] under the 1-shot and 5-shot settings on Wiki-One. However, as the pre-trained contextual information required for Nell-One is not released, the performance of RelAdapter on this dataset is unavailable for evaluation. Following RelAdapter, the comparison

---

[2]MetaR: https://github.com/AnselCmy/MetaR

[3]FAAN: https://github.com/JiaweiSheng/FAAN

[4]MetaP:https://github.com/jzystc/MetaP

[5]HiRe: https://github.com/alexhw15/HiRe

[6]GMatching: https://github.com/xwhan/One-shot-Relational-Learning

[7]FSKGC: https://github.com/chuxuzhang/AAAI2020_FSRL

[8]GANA: https://github.com/ngl567/GANA-FewShotKGC

[9]RelAdapter: https://github.com/smufang/RelAdapter

results on FB15K-One under 3-shot setting are reported in Table 3, with all baseline results sourced from RelAdapter (Ran et al., 2024). All models are implemented in PyTorch and trained on a single Tesla P100 GPU.

## B.2 Runtime Analysis

We also perform a runtime analysis comparing MoEMeta with other baselines. For a fair comparison, we evaluate MoEMeta on an RTX3090 GPU to keep consistency with baseline runtimes reported in RelAdapter (Ran et al., 2024). Table 7 reports the total runtime of meta-training and the inference time per triplet during meta-testing in seconds.

During meta-training, MoEMeta is only slightly slower than MetaR and RelAdapter, while being faster than other baselines such as GANA and FSKGC. MoEMeta requires 26,277 seconds to train on the NELL-One dataset, compared to 22,691 seconds for MetaR and 24,085 seconds for RelAdapter. The moderate increase in training time mainly comes from the two additional MLPs used in the MoE. Despite this additional overhead, MoEMeta achieves significantly better performance. For example, under the 1-shot setting on NELL-One, MoEMeta outperforms MetaR, by 28.8% MRR, 34.1% Hits@1, 27.7% Hits@5, and 24.9% Hits@10, respectively (see Table 2). For the inference during meta-testing, all methods including MoEMeta complete in milliseconds per triplet.

Table 7: Runtime comparison of meta-training and meta-testing time on Wiki-One and FB15K-One.

| Methods | Meta-training Time (seconds) | | Meta-testing Time per Triplet (seconds) | |
| --- | --- | --- | --- | --- |
| | Wiki-One | FB15K-One | Wiki-One | FB15K-One |
| GMatching | 28,941 | 19,678 | 0.009 | 0.004 |
| FSKGC | 28,742 | 20,765 | 0.013 | 0.004 |
| GANA | 35,374 | 28,167 | 0.017 | 0.006 |
| FAAN | 32,036 | 23,675 | 0.016 | 0.005 |
| HiRe | 34,257 | 27,736 | 0.036 | 0.007 |
| MetaR | 22,691 | 16,504 | 0.012 | 0.005 |
| RelAdapter | 24,085 | 17,529 | 0.045 | 0.008 |
| **MoEMeta** | 26,277 | 18,321 | 0.019 | 0.006 |

# C Detailed and Additional Related Work

For a more comprehensive discussion of related literature beyond the main paper, we provide additional related work in this section.

## C.1 Supervised KG Embedding Learning

KG embedding methods aim to learn a low-dimensional embedding space for entities and relations in KGs using input triplets as supervision. They broadly fall into two categories for KG relational learning: (1) Translation-based methods, such as TransE (Bordes et al., 2013), TransH (Wang et al., 2014), TransD (Ji et al., 2015), and TransR (Lin et al., 2015), which assume a translational relationship between entities and relations for learning a unified embedding space; and (2) Semantic matching methods, such as ComplEx (Trouillon et al., 2016), RotatE (Sun et al., 2019), and ComplEx-N3 (Lacroix et al., 2018), which improve the modeling of relational patterns in a vector or complex space. However, these methods rely on large amounts of triplets for training, and their performance degrades significantly under few-shot settings, where only a handful of training triplets are available for each relation.

## C.2 Few-Shot Learning and Meta Learning

Few-shot learning aims to train a supervised model that can generalize to new tasks with only limited amounts of training data (Fei-Fei et al., 2006). Meta-learning (Finn et al., 2017; Santoro et al., 2016; Ravi and Larochelle, 2017; Nichol and Schulman, 2018; Ren et al., 2018) has emerged as a powerful paradigm for enabling rapid learning on few-shot tasks by leveraging meta-knowledge

learned from meta-training tasks. Relative to metric-based approaches (Jake Snell and Zemel, 2017; Vinyals et al., 2016), optimization-based meta-learning methods have gained significant attention due to their strong generalization capabilities. These methods cast meta-learning into a bi-level optimization problem; the inner loop fine-tunes a base model on individual tasks, and the outer loop optimizes the base model for quick adaptation across tasks. MAML (Finn et al., 2017) is the most representative algorithm in this category, which learns global parameters as an initialization to be shared among task-specific models for fast adaptation using one or a few steps of gradient descent updates. Due to its flexible formulation, MAML has been applied to various domains, including computer vision, robotics, lifelong learning, and beyond. Subsequent variants of MAML (Finn et al., 2018; Von Oswald et al., 2021) improve upon its adaptability by optimizing model initialization or learning adaptive learning rates for faster convergence (Sun and Gao, 2023; Zou et al., 2021; Finn et al., 2018). Our work focuses on learning well-generalized meta-knowledge across tasks while integrating task-specific local adaptation to improve few-shot relational learning in KGs.

### C.3 Few-Shot Relation Learning

Few-shot relation learning (FSRL) addresses the challenges of learning with limited training examples and relational information in KGs. Current FSRL approaches can be broadly categorized into two groups: metric-learning methods and meta-learning methods. Metric-learning methods focus on learning a similarity metric between the support and query sets for adaptation. GMatching (Xiong et al., 2018) introduces one-shot relation learning, where a one-hop neighbor encoder with equal weights is used to learn entity embeddings and a matching network is designed to compare similarity between support and query entity pairs. FSKGC (Zhang et al., 2020) generalizes the setting to more shots and employs a recurrent autoencoder to aggregate few-shot instances in the support set, yet imposing an unrealistic sequential dependencies among support triplets. FAAN (Sheng et al., 2020) proposes a dynamic attention mechanism to improve the aggregation of one-hop neighbors. CSR (Huang et al., 2022) employs subgraph matching as a hypothesis testing to adapt from the support triplets to the query set. NP-FKGC (Luo et al., 2023) employs neural process to better handle distribution shifts for unseen relations.

Most mainstream FSRL methods are based on meta-learning (Finn et al., 2017), which focuses on learning an initial meta-prior from existing tasks that can generalize to new few-shot relations. MetaR (Chen et al., 2019) learns relation-specific meta-information for adaptation, by simply averaging the representations of all support triplets. GANA (Niu et al., 2021) integrates meta-learning with TransH (Wang et al., 2014) and devises a gated and attentive neighbor aggregator to address sparse neighbors for learning informative entity embeddings. HiRe (Wu et al., 2023) exploits multi-granularity relational information to learn meta-knowledge that better generalizes to unseen relations. RelAdapter (Ran et al., 2024) adopts a shallow feedforward network with a residual connection to adapt relation-meta within each target task. Additionally, pre-trained context information of related entities is used to augment entity embeddings.

Despite their great progress, the aforementioned meta-learning based methods suffer from two key limitations. First, these methods neglect common relational patterns shared across tasks when learning meta-knowledge during meta-training. Second, they fail to effectively incorporate local, task-specific information essential for fast adaptation, as global meta-knowledge may not be optimally tailored to the unique characteristics of individual tasks. Our work is thus proposed to address these critical gaps within the meta-learning framework, enabling more effective model generalization and adaptation under few-shot settings.

Our work is also related to recent graph-based and contrastive meta-learning methods such as (Liu et al., 2022) and (Liu et al., 2024). While these approaches are relevant to few-shot learning and meta-learning on node-level tasks for homogeneous graphs, they are not directly applicable to our setting, which focuses on few-shot relational learning in KGs—that centers on relation-level tasks. Our method, MoEMeta, specifically targets the key challenges of few-shot relational learning, such as the complex and diverse relational patterns in KGs. This distinction sets our work apart from general graph few-shot learning or meta-learning methods that target node-level tasks.

