# OpenReview forum: "MoEMeta: Mixture-of-Experts Meta Learning for Few-Shot Relational Learning"
_NeurIPS.cc/2025/Conference — NeurIPS 2025 poster_

### Official Review · Reviewer_fYbM · 2025-07-02

**Clarity:** 2
**Significance:** 2
**Originality:** 2
**Rating:** 4
**Confidence:** 3

**Summary:**

This paper proposes MoEMeta, a meta-learning framework for few-shot relational learning (FSRL) in knowledge graphs. The key contributions include:
1. A mixture-of-experts (MoE) module to learn shared relational prototypes across tasks, enabling global generalization.
2. A task-tailored local adaptation mechanism to refine entity/relation embeddings via dynamic projection vectors, improving task-specific adaptation.

**Questions:**

1. In the "Limitations of prior work" section, the authors only empirically identify two issues in previous methods without providing detailed referential basis. I wonder whether any existing methods have addressed these two issues or similar problems. It is expected that the discussion on limitations should incorporate more relevant references, which are essential for the paper's research motivation.
2. The "Related Work" section should not be a mere enumeration of papers, but rather reflect a summary of methodological lineages (including the problems each addresses).
3. At the methodological level of MoEMeta: 1) Do the "attentive neighbor aggregation" and the "gated and attentive neighbor aggregator" in GANA differ in implementation? 2) Does "Task-Tailored Local Adaptation" differ from TransD in implementation? The authors should clearly elaborate on the differences between each component and the referenced methods, as this is closely related to the innovativeness of the present approach.
4. The authors highlight MoEMeta's significant improvements on Nell-One. However, in Table 2, due to the unreleased information of RelAdapter, MoEMeta's notable enhancement is primarily demonstrated through comparisons with HiRe (a 2023 method). Such comparisons lack timeliness; thus, a reasonable comparison process between MoEMeta and the latest methods is required.
5. In Table 2, MoEMeta shows marginal improvements in "Hits@X" on the Wiki-one dataset, and the authors should provide corresponding analyses.

**Ethical Concerns:**

["NO or VERY MINOR ethics concerns only"]

**Final Justification:**

The authors' responses are clear and comprehensive. Therefore, I maintain my score of borderline accept.

**Limitations:**

Yes.

**Paper Formatting Concerns:**

No.

**Quality:**

3

**Strengths And Weaknesses:**

The method is straightforward and well-designed, and has been proven effective through extensive experiments. However, regarding the originality and significance of the limitations mentioned in the paper, there seems to be a lack of supporting literature or experimental results.

---

> ### Author Rebuttal · Authors · 2025-07-31
>
> > The discussion on limitations should incorporate more relevant references.
>
> Thank you for your valuable comments. Below, we provide further justification for the two limitations identified in prior work and will incorporate this discussion into the final version of our paper.
>
> First, we argue that prior FSRL methods typically learn relation-specific meta-knowledge separately for each task, but lack mechanisms to capture patterns that are globally shared across similar tasks. This issue has also been echoed in the broader few-shot and meta-learning literature. For example, works like [4] and [5] have demonstrated the benefits of modeling task similarities or intra-class commonalities within each task. However, this principle has not been systematically explored for FRSL. Our work is the first to introduce a novel MoE for explicitly learning relational prototypes and leverage these shared patterns across tasks for knowledge transfer.
>
> Second, we identify that existing meta learning-based FSRL methods typically adopt the generic Model-Agnostic Meta-Learning (MAML) framework without accounting for complex relation patterns in KGs. The KG literature has long established that relations exhibit complex, heterogeneous patterns (e.g., 1-to-N, N-to-N mappings) [6, 7], which pose a core challenge for relational learning. A standard MAML-style approach, which learns a single global initialization for all tasks, is ill-equipped to handle this structural diversity. Our work addresses this by introducing a task-tailored adaptation mechanism that explicitly uses the local context of the few-shot examples to adapt the model, a crucial step for handling the diverse nature of relations that has been overlooked for FRSL.
>
> [4] P. Zhou et al., "Task similarity aware meta learning: theory-inspired improvement on MAML." In UAI 2021, PMLR 161:23-33.
> [5] H. Li et al., "Finding task-relevant features for few-shot learning by category traversal." In CVPR 2019: 1-10.
> [6] Z. Wang et al., "Knowledge graph embedding by translating on hyperplanes." In AAAI 2014: 1112-1119.
> [7] G. Ji et al., "Knowledge graph embedding via dynamic mapping matrix." In ACL/IJCNLP 2015: 687-696.
>
> > The "Related Work" section should reflect a summary of methodological lineages (including the problems each addresses).
>
> Thank you for the valuable feedback. In our Related Work section, we have aimed not simply to list prior papers but to summarize the key limitations of existing methods, which clearly motivates the research gaps that our work targets. The discussion around methodological lineages is kept concise in the main paper mainly due to space limit. Yet, we have provided a more detailed discussion in Appendix C.3. We will further strengthen the Related Work section to ensure the narrative clearly emphasizes these aspects.
>
> > At the methodological level of MoEMeta: 1) Do the "attentive neighbor aggregation" and the "gated and attentive neighbor aggregator" in GANA differ in implementation? 2) Does "Task-Tailored Local Adaptation" differ from TransD in implementation?
>
> Thank you for the insightful questions. We address both points as follows:
>
> 1). **Attentive neighbor aggregation vs. GANA’s gated and attentive aggregator:** Our attentive neighbor aggregation module is designed as a lightweight mechanism to incorporate local entity context, using a simple attention scheme over one-hop neighbors. This component is not the focus of our work; thus, we intentionally adopt a simple design to ensure that improvements stem from our core contributions (i.e., the MoE-based meta-knowledge and task-specific adaptation). In contrast, GANA's main contribution lies in designing a more sophisticated gated and attentive aggregation module with multiple gating and transformation layers to better handle sparse neighbours. We do not use the gated structure or GANA’s architecture in our implementation.
>
> 2). **Task-Tailored Local Adaptation vs. TransD:** Our task-specific adaptation mechanism is inspired by TransD in using projections to model the interaction between entities and relations. However, there is a key difference in how these projection vectors are treated:
>    - In TransD, the projection vectors are learned globally as part of the shared model parameters, and remain fixed across tasks or relations once trained.
>    - In contrast, our projection vectors are randomly initialized for each task and optimized only within that task, allowing for more flexible and task-specific adaptation in the few-shot setting. This design enables MoEMeta to better capture task-level semantics without relying on global sharing, which may not generalize well to unseen relations.
>
> > A reasonable comparison process between MoEMeta and the latest methods is required.
>
> Thank you for your comment. We would like to clarify that RelAdapter is a recent 2024 method, and we have included comparative results with it on both FB15K-One and Wiki-One, as shown in Table 2 and Table 3. As RelAdapter does not report results on NELL-One, we compare MoEMeta against the strongest available baseline on NELL-One, HiRe. Despite being a 2023 method, HiRe remains a competitive baseline in meta-learning based FSRL literature, especially on NELL-One.
>
> We acknowledge that meta-learning based FSRL has seen limited updates in recent years. We will continue to track and include future methods as they become available, and are prepared to include additional comparisons in an updated version of the paper if new implementations or results are released.
>
> > Marginal improvements in "Hits@X" on the Wiki-one dataset.
>
> Thank you for valuable suggestion. We agree that the improvements of MoEMeta on the Wiki-One dataset are less pronounced compared to NELL-One. A key reason lies in the nature of relation representations in Wiki-One: In Wiki-One, relations are represented using non-semantic identifiers that lack natural language descriptions. In contrast, NELL-One uses textual relation names, which often exhibit explicit semantic regularities and correlations. Since MoEMeta is designed to model and leverage commonalities across relations through expert sharing, its advantage is more evident when relation semantics can be meaningfully captured. We will add this analysis to the revised version to clarify the differences across datasets.

---

> > ### Comment · Reviewer_fYbM · 2025-08-05
> > **Official Comment by Reviewer fYbM**
> >
> > The authors' responses are clear and comprehensive.
> >
> > However, I have another suggestion: please provide a clear and detailed explanation of "Attentive Neighbor Aggregation" and "Task-Tailored Local Adaptation" in the main text, as raised in the third question.
> >
> > Ultimately, I maintain my score of borderline accept.

---

> > > ### Author Response · Authors · 2025-08-07
> > >
> > > We thank the reviewer for this constructive suggestion. In the final version, we will incorporate the additional explanations from our rebuttal and make further clarifications to provide a clearer and more detailed description of the two components.

---

### Official Review · Reviewer_oYNg · 2025-07-02

**Clarity:** 3
**Significance:** 3
**Originality:** 3
**Rating:** 4
**Confidence:** 3

**Summary:**

This paper presents MoEMeta, a novel meta-learning framework designed to improve few-shot relational learning in knowledge graphs (KGs). Few-shot KG relational learning is challenging due to the limited number of examples for some relations. Most existing solutions follow a meta-learning approach, but they often fail to capture common relational patterns across related tasks and overlook important task-specific information. MoEMeta addresses these issues with two key innovations: a mixture-of-experts (MoE) model that learns relational prototypes shared across tasks and a task-tailored adaptation mechanism for better local adaptation to task-specific contexts. By balancing global generalization and local adaptability, MoEMeta promises improved few-shot relational learning. The paper demonstrates the effectiveness of MoEMeta through extensive experiments and ablation studies on three KG benchmarks, where it outperforms existing state-of-the-art methods.

**Questions:**

see above

**Ethical Concerns:**

["NO or VERY MINOR ethics concerns only"]

**Final Justification:**

The authors' responses are clear. I maintain my borderline accept score.

**Limitations:**

see above

**Quality:**

3

**Strengths And Weaknesses:**

## Paper Strength

(1) The paper introduces a new meta-learning framework, MoEMeta, which successfully addresses two main limitations in current few-shot relational learning methods by emphasizing both global and local adaptability.

(2) MoEMeta incorporates a mixture-of-experts model, which is a creative way to capture relational prototypes shared across tasks and thus enhances model generalization in few-shot scenarios.

(3) The task-tailored adaptation mechanism adds a valuable dimension for capturing the unique task-specific information, offering superior local adaptation capabilities compared to existing methods.


## Paper Weakness
(1) Figure 1 lacks clarity – The current illustration does not clearly or sufficiently convey the proposed method or its components. It would be helpful to improve the figure's design and add more detailed annotations to enhance understanding.

(2) Incremental contribution – The core idea of the paper appears to be a straightforward combination of Mixture-of-Experts (MoE) with task-specific modules. While the integration is reasonable, it does not offer significant innovation beyond existing approaches.

---

> ### Author Rebuttal · Authors · 2025-07-31
>
> > Figure 1 lacks clarity.
>
> Thank you for your advice. The figure layout is constrained by space limit. we will revise Figure 1 to improve clarity and provide more details to illustrate the main components of the proposed method.
>
> > Incremental contribution. While the integration is reasonable, it does not offer significant innovation beyond existing approaches.
>
> We understand that our work might initially appear as a combination of existing components. However, we respectfully argue that our technical novelty lies in proposing a principled meta-learning framework to solve a non-trivial, previously unaddressed challenge in FSRL: how to disentangle globally shared patterns across tasks from local task specifics to facilitate fast adaptation to unseen tasks under few-shot settings.
>
> We would like to reiterate the novel aspects of our work. First, we have correctly identified valid research limitations in the existing few-shot relational learning. These strongly motivate a compelling need for novel methodology to address these limitations. Second, we have proposed two key components to explicitly address these important research gaps:
>
> - **A novel MoE for learning shared relational prototypes:** Unlike existing methods that learn meta-knowledge only within individual tasks, our MoE-based meta-learner is explicitly designed to capture common relational patterns shared across similar tasks for learning generalizable meta-representations. This is a novel formulation that explicitly models the compositional nature of relations in KGs, but also a clear distinction from previous work.
>
> - **A novel task-tailored adaptation mechanism**: Unlike prior FSRL methods that relies on a single set of global parameters for initialization (a uniform MAML approach). Our work introduces a task-tailored adaptation mechanism that fine-tunes the meta-representations and entity embeddings tailored to the unique structural context of each task. Our adaptation is a novel, lightweight solution, specifically designed to capture the diverse interactions between entities and relations in individual tasks.
>
> Therefore, our key innovation is a novel unified framework that enables global generalization across similar tasks and local adaptability. Our experimental evaluation demonstrates the state-of-the-art performance of our proposed solution, affirming the novelty of our contributions. We will revise the paper to clarify our contributions more explicitly.

---

> > ### Comment · Reviewer_oYNg · 2025-08-06
> >
> > I maintain my score of borderline accept.

---

### Official Review · Reviewer_QaZM · 2025-07-02

**Clarity:** 3
**Significance:** 3
**Originality:** 2
**Rating:** 4
**Confidence:** 4

**Summary:**

This paper proposes a MoEMeta model that introduces MoE mechanism into meta learning-based few-shot relational learning framework. Experimental results on three few-shot knowledge graph completion datasets illustrate the effectiveness of MoEMeta compared with other baseline models. However, the novelty of this paper should be improved and declared more clearly.

**Questions:**

1. How the proposed model performs on traditional datasets such as FB15K237 and WN18RR?
2. Why the amount of experts is set as 32 on all the three datasets?

**Ethical Concerns:**

["NO or VERY MINOR ethics concerns only"]

**Final Justification:**

Some of the concerns have been addressed.

**Limitations:**

The number of model parameters would increase significantly due to the utilization of MoE mechanism. It is suggested to conduct the analysis of the model's complexity.

**Paper Formatting Concerns:**

N/A.

**Quality:**

3

**Strengths And Weaknesses:**

Strengths:
1. This paper is well-written and easy to follow.
2. The authors develop a few-shot knowledge graph completion model with the widely-used MoE strategy to model various relation patterns.
3. The experimental results illustrate the effectiveness of MoEMeta.

Weaknesses:
1. This work is the extension of some existing models such as MetaR and GANA. However, simply introducing the MoE strategy into meta learning-based few-shot relational learning is not sufficient to demonstrate the novelty of the proposed approach.
2. The lack of experimental results on traditional datasets such as FB15K237 and WN18RR makes it difficult to demonstrate the generalization of the developed model.
3. Since the semantics and the amount of relations vary significantly across different datasets, setting the same number of experts uniformly is not very reasonable. The number of experts should be dynamically adjusted based on the specific characteristics of each dataset.

---

> ### Author Rebuttal · Authors · 2025-07-31
>
> We respectfully reiterate that MoEMeta is **NOT** a simple extension of existing methods such as MetaR or GANA. Instead, we have proposed a novel unified framework that enables both generalization across similar tasks and fine-grained local adaptation. Below, we would like to clarify the technical novelty of our approach, which addresses two fundamental limitations of prior methods.
>
> - **Capturing shared knowledge across tasks**: Unlike existing methods that learn meta-knowledge only within individual tasks, our MoE-based framework is explicitly designed to capture common relational patterns shared across similar tasks. This is a key departure from previous work and allows for better generalization.
>
> - **Task-tailored adaptation**: Prior methods often use a single set of global parameters for initialization (a uniform MAML approach). Our work introduces a task-tailored adaptation mechanism that fine-tunes the shared knowledge using the unique structural context of each task.
>
> Importantly, our work is **NOT** a simple introduction of MoE into few-shot relational learning. It is a new framework carefully designed to disentangle what to generalize (globally shared patterns across tasks) from what to adapt (local task specifics) --- a distinction that prior work does not address. We believe that this is not only novel, but also opens up a new research direction in few-shot relational learning, encouraging future work to explore structured task relationships, rather than treating each task as i.i.d.
>
> > How the proposed model performs on traditional datasets such as FB15K237 and WN18RR?
>
> Thanks for your question. We have provided experimental results on **FB15K-One**, a few-shot version derived from FB15K237, to evaluate the proposed MoEMeta. In the literature on few-shot relational learning, **NELL-One** and **Wiki-One** are commonly used benchmarks, whereas **WN18RR** is rarely adopted. One key reason is that WN18RR contains only **11 relations**, making it unsuitable for constructing meaningful few-shot tasks with sufficient diversity.
>
> > Why the amount of experts is set as 32 on all the three datasets?
>
> Thank you for this thoughtful question. To clarify, we did not uniformly fix the number of experts across different datasets. Instead, we treated the number of experts as a key hyperparameter and performed a sensitive analysis, as detailed in Section 5.4. On each dataset, we tuned the number of experts in the range of $\{16, 32, 64\}$ based on practical guidance from the number of relations and task diversity on that dataset. Our empirical results reveal a consistent finding: performance on all three datasets peaks or plateaus when the number of experts was set to 32, striking a good balance between model capacity and generalization.
>
> > It is suggested to conduct the analysis of the model's complexity.
>
> Thanks for this valuable suggestion. We would like to draw your attention to our detailed analysis of model complexity and runtime in the appendix.
>
> - **Complexity Analysis (Appendix A.2)**: We analyze the complexity of all MoEMeta components with respect to the number of parameters and the number of multiplication operations in each epoch. The table below summarizes this analysis.
>
> | Component           | # Parameters      | # Operations       |
> |---------------------|-------------------|---------------------|
> | Neighbor aggregator | $\mathcal{O}(d^2)$ | $\mathcal{O}(nkd^2)$ |
> | MoE                 | $\mathcal{O}(d^2)$ | $\mathcal{O}(nd^2)$  |
> | Local adaptation    | $\mathcal{O}(d)$   | $\mathcal{O}(d)$     |
> | **Total**           | $\mathcal{O}(d^2)$ | $\mathcal{O}(nkd^2)$ |
>
>
> The MoE's parameter overhead of $\mathcal{O}(d^2)$ ($d$ is the embedding dimension) is on par with the complexity of the neighbor aggregator, which is an essential component in most SOTA methods. Thus, this additional parameter overhead is marginal relative to the model's overall complexity.
>
> - **Runtime Performance (Appendix B.2)**: We provide a runtime comparison with baselines in Appendix B.2. Table 7 reports the total runtime for meta-training and the inference time per triplet during meta-testing in seconds. The table below summarizes this comparison.
>
> | Model              | Meta-training Wiki-One (second) | Meta-training FB15K-One (second) | Meta-testing per triplet on Wiki-One (second) | Meta-testing per triplet on FB15K-One (second) |
> | ------------------ | ------------------------------- | -------------------------------- | --------------------------------------------- | ---------------------------------------------- |
> | GMatching          | 28941                           | 19678                            | 0.009                                         | 0.004                                          |
> | FSRL               | 28742                           | 20765                            | 0.013                                         | 0.004                                          |
> | GANA               | 35374                           | 28167                            | 0.017                                         | 0.006                                          |
> | FAAN               | 32036                           | 23675                            | 0.016                                         | 0.005                                          |
> | HiRe               | 34257                           | 27736                            | 0.036                                         | 0.007                                          |
> | MetaR              | 22691                           | 16504                            | 0.012                                         | 0.005                                          |
> | RelAdapter         | 24085                           | 17529                            | 0.045                                         | 0.008                                          |
> | **MoEMeta (ours)** | **26277**                       | **18321**                        | **0.019**                                     | **0.006**                                      |
>
> During meta-training, MoEMeta is only slightly slower than MetaR and RelAdapter, but faster than other baselines such as GANA and FSKGC. On NELL-One, MoEMeta requires 26,277 seconds to train, compared to 22,691 seconds for MetaR and 24,085 seconds for RelAdapter. This moderate increase in training time mainly comes from the two MLPs in the MoE. During meta-testing, MoEMeta, like all baselines, is highly efficient, with an inference time of milliseconds per triplet.

---

> > ### Comment · Reviewer_QaZM · 2025-08-06
> >
> > Some of the concerns have been addressed. Thus, I raise the score. Introducing MoE into meta learning-based Few-shot KGC  framework is really an interesting idea but the clarification of the novelty is suggested to be further enhanced.

---

> > > ### Author Response · Authors · 2025-08-06
> > >
> > > Thank you for your valuable suggestion and for acknowledging the interesting idea of our work.
> > >
> > > We are pleased that our rebuttal has addressed your main concerns. We will revise the final version of the paper to more clearly elaborate on the technical novelty and contribution of our work.

---

### Official Review · Reviewer_XdzE · 2025-07-06

**Clarity:** 3
**Significance:** 2
**Originality:** 2
**Rating:** 4
**Confidence:** 2

**Summary:**

This paper proposes a new meta-learning method for few-shot relational learning. Specifically, the authors focus on task-specific modulation mechanism for both meta-knowledge representation and task-specific adaptation. For meta-knowledge part, each entity is represented by aggregating its neighborhood information, which is processed with the attention-like mechanism. Then, this entity representation is used for gating the MoE mechanism responsible for predicting the correct relation given the support set. For the task-specific adaptation part, the authors propose to task-specifically learn local projections for the embeddings of head/tail entities and relations. All the meta-parameters are learned through MAML-like algorithm. The experimental results demonstrate the effectivenss of the proposed method on some popular benchmark datasets, outperforming all the baselines considered in this paper.

**Questions:**

Overall, the results seem appealing and I agree with the proposed method. Each component has carefully been demonstrated to be effective as well. The only concern is novelty, which I'm not sure. I'm willing to increase my score (up to 1) if my concern can be addressed properly.

Another suggestion is if the proposed method can be tested on more baselines and more public datasets. The most recent baselines are from two years ago, which makes me skeptical about the true SOTA-ness. Also, I'm not sure whether the two datasets, Nell-One and Wiki-One, are really sufficient for generalizing the effectiveness of the proposed method to other unseen datasets.

**Ethical Concerns:**

["NO or VERY MINOR ethics concerns only"]

**Final Justification:**

Authors' rebuttal was satisfactory

**Limitations:**

Limitations have not been addressed in this paper.

**Quality:**

3

**Strengths And Weaknesses:**

Strengths
- Paper is well written and well organized.
- The proposed method is simple, intuitive and makes sense.
- Each individual component is shown effective by the ablation study.
- The proposed method achieves superior performance to the existing baselines.

Weaknesses
- While I'm very familiar with meta-learning, I admit that I'm not an expert in relational learning area. My overall impression is that the proposed method intuitively makes sense. However, it also feels like a little bit straightforward in terms of task-specific modulation. In meta-learning community, such task-specific modulation has long been studied since 2018. It's a bit surprising that no prior work has studied such task-specific modulation in the context of relational learning area. Since I'm not following any work in relational learning area, I need to consult with other reviewers and AC.
- More specifically, it's surprising that no prior work has considered aggregating the neighbour information when processing each entity or relation. Task-specific MoE-like gating is not a new idea in meta-learning as well. The only difference seems whether it's for relational learning or not. Task-specific adaption is also not a new idea if we speak broadly, but the details seem novel. I deeply understand the importance of the task-specific modulation in meta-learning, but I also wonder why the previous literature of relational learning have not explored such a natural direction. Again, I'm not an expert in this area, so I politely ask this to both the authors and the other reviewers.

---

> ### Author Rebuttal · Authors · 2025-07-31
>
> We sincerely thank the reviewer for providing supportive comments and acknowledging the strengths of our work. We respond to your questions and provide clarifications as follows:
>
> > In meta-learning community, such task-specific modulation has long been studied since 2018. It's a bit surprising that no prior work has studied such task-specific modulation in the context of relational learning area.
>
> We agree with the reviewer that while task-specific modulation has been well studied in meta learning, its application to relational learning has been surprisingly limited. We believe that this gap exists precisely because a direct application of modulation from general meta-learning is inapplicable or insufficient for the structured, non-i.i.d. domain of knowledge graphs (KGs) for few-shot relational learning (FSRL), where only a handful of training triplets are available per task. Our work is thus proposed to fill this critical gap by proposing a novel unified framework that enables both generalization across similar tasks and fine-grained local adaptation.
>
> To our best knowledge, RelAdapter (published in 2024) is the only baseline using task-specific adaptation. However, its simple approach--using an MLP to generate a single adaptation vector--fails to account for diverse interactions between entities and relations tailored to individual tasks. Our proposed approach addresses this limitation through explicitly projecting entity and relation embeddings into a task-specific subspace, offering a more principled solution for task-specific adaptability.
>
> > More specifically, it's surprising that no prior work has considered aggregating the neighbor information when processing each entity or relation. Task-specific MoE-like gating is not a new idea in meta-learning as well. The only difference seems whether it's for relational learning or not. Task-specific adaption is also not a new idea if we speak broadly, but the details seem novel.
>
> Thank you for this insightful feedback. We would like to clarify the distinct contributions of our work by addressing each point.
>
> - **On neighbor aggregation**: We clarify that neighbor aggregation is not novel; it is an essential component used by SOTA methods. Our work builds upon this standard practice, which is why we did not claim it as a contribution.
>
> - **Technical novelty of our work**: We agree that the high-level concepts of task-specific modulation and MoE are not new in isolation in general meta-learning. Our core technical novelty lies in designing a principled framework that solves a critical, non-trivial challenge unaddressed by prior FSRL work: **How to disentangle shared, global knowledge from task-specific, local context to enable effective generalisability and fast adaptation under few-shot settings.**
>
> Our MoEMeta framework provides a principled solution to this challenge:
> - **Global knowledge (what to generalize)**: This is achieved by our MoE-based relational prototypes that capture **common relational patterns shared across similar tasks** for learning relation-meta representations. Unlike Meta-DMoE [1] where each expert is specialized for a specific task domain, our MoE is designed to learn common relational patterns shared across tasks for better generalization. Our experts are trained to learn distinct relational prototypes. The gating network then learns to represent each new, unseen relation as a weighted combination of these shared, fundamental prototypes. This is a novel formulation that explicitly models the compositional nature of relations in KGs.
>
> [1] Zhong, Tao, et al. "Meta-DMoE: Adapting to domain shift by meta-distillation from mixture-of-experts." NeurIPS 2022: 22243-22257.
>
> - **Local knowledge (what to adapt)**: This is achieved by our task-tailored adaptation mechanism that leverages task-specific structural context--unique to each task--to fine-tune relation-meta and entity embeddings. As the reviewer noted, "the details seem novel." and this distinction is crucial. Unlike general task-specific modulation methods [2, 3], which focus on predicting task mode to modulate the meta-learned prior parameters, our adaptation is a lightweight solution operating at the embedding level, specifically designed to capture the diverse interactions between entities and relations in individual tasks. For each task, we locally learn three randomly initialized projection vectors, allowing for highly specific and efficient adaptation.
>
> [2] R. Vuorio et al., Multimodal Model-Agnostic Meta-Learning via Task-Aware Modulation, NeurIPS 2019: 14632-14644.
>
> [3] Mi. Abdollahzadeh et al., Revisit Multimodal Meta-Learning through the Lens of Multi-Task Learning, NeurIPS 2021: 1-12.
>
> This dual approach forms a novel unified framework that enables both generalization across similar tasks and fine-grained local adaptation. We will revise the paper to make these points more explicit and better contextualize our contributions with respect to the broader meta-learning literature.
>
> > More baselines and more public datasets.
>
> We appreciate your valuable suggestion. We aimed to provide a thorough comparison against the most relevant and current methods. To clarify, our evaluation includes RelAdapter (published in 2024), which is, to our best knowledge, the latest SOTA baseline within the meta-learning paradigm for this task.
>
> For fair comparison, our use of Nell-One and Wiki-One is consistent with the standard evaluation protocol used by the majority of prior work in few-shot relational learning. To further validate the generalizability of our method, we have also expanded our experiments to include a third dataset, FB15K-One, adopting the same setup introduced by the SOTA method, RelAdapter. The results are reported in Table 3.

---

> > ### Comment · Reviewer_XdzE · 2025-08-05
> >
> > Thanks for the rebuttal. Most of my concerns have been resolved, thus I increase my score.

---

> > > ### Author Response · Authors · 2025-08-06
> > >
> > > Thank you for your thoughtful feedback again and for taking the time to review our rebuttal.
> > >
> > > We are glad to hear that we were able to resolve your concerns, and we appreciate your updated evaluation. We will incorporate key points from our rebuttal to further highlight the novelty of our work in the final version.

---

### Note · Authors · 2025-08-12

Dear Reviewers, Area Chairs, and Senior Area Chairs,

We sincerely thank the reviewers for their constructive comments and active engagement. The discussion period was invaluable, allowing us to directly resolve initial concerns regarding technical novelty of our work, its comparison with SOTA baselines, and clarifications on the choice of benchmark datasets and complexity analysis detailed in the appendix.

We appreciate that our clarifications resulted in two reviewers increasing their scores, leading to a consensus of support. We believe this positive trajectory affirms that the paper's primary concerns have been successfully resolved.

We are committed to incorporating all of the reviewers' valuable suggestions to further strengthen the final manuscript, should the paper be accepted. Thank you for your time and consideration.


Warm regards,

Authors of Submission 3740

---

### Decision · Program_Chairs · 2025-09-17

**Decision:**

Accept (poster)

**Comment:**

This paper introduces a meta-learning approach for few-shot relational learning that incorporates task-specific modulation in both meta-knowledge representation and task-level adaptation. For meta-knowledge, entities are represented by aggregating neighborhood information and these representations are used to gate a mixture-of-experts (MoE) module for relation prediction given a support set. For task-specific adaptation, the method learns local projection functions for head and tail entity embeddings as well as relation embeddings. All meta-parameters are optimized via a MAML-style algorithm. Experiments on standard benchmark datasets show improved performance from the proposed method.

After discussion, all reviewers recommended acceptance, though some limitations and concerns remain. The use of MoE is an interesting contribution though the general approaches adopted in this work has been widely used (e.g., task-specific modulation).